# Retrospective Analysis of Prognostic Factors in Pediatric Patients with Adrenocortical Tumor from Unique Tertiary Center with Long-Term Follow-Up

**DOI:** 10.3390/jcm11226641

**Published:** 2022-11-09

**Authors:** Fernanda S. Bachega, Caio V. Suartz, Madson Q. Almeida, Vania B. Brondani, Helaine L. S. Charchar, Amanda M. F. Lacombe, Sebastião N. Martins-Filho, Iberê C. Soares, Maria Claudia N. Zerbini, Francisco T. Dénes, Berenice Mendonca, Roberto I. Lopes, Ana Claudia Latronico, Maria Candida B. V. Fragoso

**Affiliations:** 1Unidade de Suprarrenal, Laboratório de Hormônios e Genética Molecular LIM/42, Disciplina de Endocrinologia e Metabologia, Hospital das Clínicas, Faculdade de Medicina da Universidade de São Paulo, São Paulo 05466-040, SP, Brazil; 2Divisão de Urologia, Departamento de Cirurgia da Faculdade de Medicina da Universidade de São Paulo, São Paulo 1964-2007, SP, Brazil; 3Departamento de Patologia, Faculdade de Medicina da Universidade de São Paulo, São Paulo 01246-903, SP, Brazil

**Keywords:** pediatric adrenocortical tumor, prognostic factors, pediatric cancer, *TP53*

## Abstract

Pediatric adrenocortical tumors (PACTs) represent rare causes of malignancies. However, the south/southeast regions of Brazil are known to have a high incidence of PACTs because of the founder effect associated with a germline pathogenic variant of tumor suppressor gene *TP53*. We aimed to retrospectively analyze the types of variables among hormone production, radiological imaging, tumor staging, histological and genetic features that were associated with the occurrence of malignancy in 95 patients (71% females) with PACTs from a unique center. The worst prognosis was associated with those aged > 3 years (*p* < 0.05), high serum levels of 11-desoxicortisol (*p* < 0.001), tumor weight ≥ 200 g (*p* < 0.001), tumor size ≥ 5 cm (*p* < 0.05), Weiss score ≥ 5 (*p* < 0.05), Wieneke index ≥ 3 (*p* < 0.001) and Ki67 ≥ 15% (*p* < 0.05). Furthermore, patients with MacFarlane stage IV had an overall survival rate almost two times shorter than patients with other stages (*p* < 0.001). Additionally, the subtractions of *BUB1B*-*PINK1* (<6.95) expression (*p* < 0.05) and *IGF-IR* overexpression (*p* = 0.0001) were associated with malignant behavior. These results helped identify patients who are likely to have an aggressive course; further multicenter prospective studies are required to confirm our results. In conclusion, PACTs with these patterns of prognostic factors could be treated using an adjuvant approach that may improve the overall survival in such patients.

## 1. Introduction

Adrenocortical tumors (ACTs) comprise benign adrenocortical adenomas (ACAs) and malignant adrenocortical carcinomas (ACCs). These tumors can develop at any age regardless of sex. However, they typically occur during the first and fourth decades of life, characterizing the bimodal incidence [1], with a female-to-male ratio of 2.3:1 [2].

The estimated global prevalence of pediatric adrenocortical tumors (PACTs) is 0.5 cases/million population [3]. However, ACC is rare and accounts for 0.2% of all childhood malignancies, with considerable variability in incidence rates in different geographical areas [4]. The incidence of PACTs in the south and southeast regions of Brazil is known to be 15–18 times higher than in other areas [2,5], with an estimated prevalence of 3.4–4.2 cases/million population [6]. This high incidence rate has been attributed to a specific germline pathogenic variant at codon 337 (c.1010G > A, p. R337H) of p53, characterizing a founder effect [7,8]. The pathogenic variant p.R337H is located outside the DNA-binding domains. The arginine residue at codon 337 is part of an alpha-helix motif involved in p53 oligomerization, and the replacement of arginine with histidine disrupts oligomerization in a pH-dependent manner, causing the domain to fail to oligomerize under slightly elevated pH conditions. In pediatric patients, this germline variant indicates a hereditary characteristic of predisposition to several neoplasms (such as choroid plexus tumor, leukemia, etc.); however, differently than adults, pediatric patients with ACTs are not generally associated with a worst prognosis [9].

Li–Fraumeni/Li–Fraumeni-like syndromes (LFS/LFL) are characterized by early-onset familial cancer with autosomal-dominant predispositions [9,10,11]. Cases of LFS/LFL account for 3–10% of all cases of ACC [9]. In contrast to adults with ACC, the presence of this germline pathogenic variant of *TP53* in pediatric patients is not correlated with poor prognoses [12]. Clinical variability is commonly observed in LFS, causing phenotypic heterogeneity among different families affected by the same pathogenic variant in *TP53* gene. This clinical spectrum may be due to the influence of the modifier genes [13].

We previously analyzed the involvement of other genes, such as *MLH1*, *MSH6*, *ATRX*, *ZNRF3*, *XAF1*, *BUB1B-PINK1*, and *IGF-IR* expression, in the occurrence of PACTs [13,14]. These findings partly explained the varied genotypes, which can be linked to aggressive outcomes. Additionally, the role of environmental or other genetic factors in the development of PACTs and their effects on the biological behavior of these tumors is not well characterized. It is important to identify prognostic factors in order to guide clinical practice, facilitate optimal therapeutic planning, and improve treatment outcomes.

In this retrospective study, we aimed to identify the epidemiological, clinicopathological and genetic factors associated with the development of metastasis or fatal outcomes in a cohort of patients diagnosed with PACT at a unique tertiary care center in Brazil based on a long-term follow-up study. Additionally, we compiled published data from this same care center to analyze all aforementioned criteria simultaneously, enhancing prognostic factors in the entire cohort.

## 2. Materials and Methods

This retrospective study reviewed 95 medical records of pediatric patients diagnosed with ACT between 1977 to 2021 at the Department of Endocrinology and Metabolism of the Adrenal Unit and Department of Surgical Urology at the Hospital das Clínicas of São Paulo University. Informed consent was obtained from all patients or their legal guardians.

Clinical presentation, tumor hormonal secretion, radiological imaging, histological features (immunohistochemical Ki67% and p53), and genetic data (*TP53*, *MLH1*, *MSH6*, *ATRX*, *ZNRF3*, *XAF1*, *BUB1-PINK*, *IGF-IR*) were included in the analysis.

All patients aged up to 18 years, according to Brazil’s Statue of the Child and Adolescent, underwent uneventful laparoscopic or open surgery with a histological diagnosis of ACT confirmed [15]. Based on previous study [6], the patients were categorized into three age-groups: group 1: ≤3 years of age; group 2: 3 < age ≤ 12 years of age; group 3: 12 < age ≤ 18 years of age. The groups were further subdivided into the following subgroups: metastatic and non-metastatic.

The histological and immunohistochemical reports were reviewed by three expert pathologists (SNMF; ICS; MCNZ) to confirm the tumor features and indices such as Weiss/modified Weiss score and Wieneke index and Ki67% immunostaining. Furthermore, we analyzed the tumor staging by TNM and by McFarlane and MacFarlane/Sullivan. Overall survival (OS), disease-free survival (DFS) and recurrence-free survival (RFS) curves were estimated using the Kaplan-Meier method [16].

The immunohistochemical analysis for Ki67 was performed in the most highly mitotic or most cellular blocks/slides from each tumor: 4 μm sections were cut, deparaffinized and rehydrated. Antigen retrieval was achieved using 10 mM pH 6.0 citrate buffer solution for 40 min in a steamer at 95 °C. After the peroxidase block, the sections were incubated overnight with primary antibody (mouse monoclonal antibody MIB1, 1:100 dilution, DAKO, Glostrup, Denmark) at 4 °C. The Novolink Polymer Detection System (Vision Biosystems TM, Sheffield, UK), followed by DAB-solution revelation (3,3′-diaminobenzidine tetrahydrochloride and dimethyl sulfoxide—Sigma, Burlington, MA, USA) performed the signal amplification. Slides were counterstained with hematoxylin, dehydrated, and mounted with Entellan (Merck, Germany). A stained section of a lymph node was used as an external positive control. Furthermore, the endothelial cells and reactive lymphocytes within the tumors were used as internal positive controls [17]. Ki67 evaluation was visually performed at the microscope in increments of 10%, in a range of 0 to 100%. Data for melan A, SF1, inhibin and p53 were collected in immunohistochemical reports when available and categorized as negative or positive (any positivity).

The description of genetic materials and methods can be found in the Appendix A.

Categorical variables were presented as absolute (*n*) and relative (%) frequency distributions. The main summary measures for quantitative variables used are mean, standard deviation, median, minimum and maximum values. Patients who did not present the event of interest (progression/death) throughout the study were censored. Survival curves were estimated using the Kaplan-Meier method, and the between-group differences in survival outcomes were assessed using log-rank test. In addition, Cox’s proportional hazards model was used to describe the relationship of variables with time to progression/death [18], and the results presented as hazard ratios (HR) with confidence intervals (CI 95%). Due to the small sample size within each age-group and the low frequency of certain events, the multivariate model was not adjusted. In all fitted models, the assumption of proportionality was evaluated using Schoenfeld residuals [19]. In all analyses, there is evidence that the effect of covariates was constant over time, thus justifying the use of the Cox model. The same analyses were conducted for the general sample, comparing DFS and OS in relation to the age-group.

## 3. Results

### 3.1. Epidemiologic Features: Age, Gender, and Geographic Origin from Brazil

The mean age of all patients at diagnosis was 5.8 years of age (range 0.58—18 years of age); 71.5% were female (F:M ratio 2.5:1). The follow-up ranged from 27 to 408 months. The demographic characteristics of the study population are summarized in Table 1. Three groups were defined: group 1: ≤3 years of age; group 2: 3 < age ≤ 12 years of age; group 3: 12 < age ≤ 18 years of age.

The F:M ratio in groups 1, 2, and 3 were 2.6:1; 2.5:1, and 5.6:1, respectively. Although the proportion of females was higher than males in all groups, the gender showed no significant association with prognosis (*p* = 0.09; HR: 4.2; CI 95% 0.78–23.3).

Although the youngest group (group 1) accounted for the majority of patients, it had the lowest percentage of metastatic patients at diagnosis (11.3%). Group 2, with an intermediate value between the other two groups, had 35.7% metastatic patients. The adolescent group (group 3) had the highest percentage of metastatic patients at diagnosis (47.4%) (Figure 1). The patients aged > 3 years were more than twice as likely to develop a malignant evolution, mainly in the adolescent group (*p* = 0.03; HR: 2.5; CI 95% 1.07–5.8).

### 3.2. Clinical Presentation

Group 1 had the highest percentage of patients with isolated virilizing syndromes (VS; 70%); group 2 had 35.7%, whereas group 3 had the lowest percentage of these clinical presentations (31.6%). The mixed clinical presentation (VS and Cushing’s syndrome (CS)) was observed in 30%, 57.1%, and 26.3% of patients in groups 1, 2, and 3, respectively. None of the patients in group 1 had isolated CS in contrast to 3.6% of patients in group 2; group 3 had the highest percentage of isolated CS, affecting 24% of patients. Feminizing syndrome (FS) was present in groups 2 and 3, affecting 3.6% and 18% of patients in these groups, respectively. Nevertheless, the clinical presentation showed no significant association with prognosis in any of the groups (*p* = 0.057; HR 0.042; CI 95% 0.002–1.1).

Figure 2 shows some of the main clinical features of patients with PACTs. The majority of patients with VS presented with peripheral precocious puberty, characterized by pubic hair (87.3%), macrogenitosomy or clitoromegaly (67.1%), linear growth acceleration (64.9%), advanced bone age (65%), hirsutism (87.3%), deepened voice (29.4%), and muscle hypertrophy (67%). The most frequent presenting features of CS were weight gain (81%), blood hypertension (36.4%), buffalo hump (7.1%), plethora/moon face (21.1%), wide purple stretch marks (23%), and a decrease in linear growth (35.8%). Only two patients presented an estrogen-secreting tumor (FS) associated with gynecomastia and severely advanced bone age.

### 3.3. Hormonal Data

Hormonal data are presented in Table 2. Group 3 had the highest mean level of 11-deoxycortisol. In addition, in all the groups, patients with higher serum levels of 11-deoxycortisol were three times more likely to develop metastasis (*p* < 0.001; HR 2.9; CI 95% 0.9–1.2), associating 11-deoxycortisol with worse outcomes.

The mean levels of androgen hormones (dehydroepiandrosterone sulfate (S-DHEA), dehydroepiandrosterone [20], testosterone, and free testosterone) in younger patients were higher than those in older patients. However, the levels of androgen hormones showed no significant association with a worse prognosis (*p* > 0.05). Additionally, serum testosterone levels were higher in children with pure virilization symptoms compared to those with mixed syndromes (VS + CS).

### 3.4. Radiological Data

The preferred imaging modalities used to verify, characterize, and localize adrenal tumors and staging (e.g., the presence or absence of metastasis) were abdominal magnetic resonance imaging, computed tomography (CT) for lung, and PET/CT FDG scans.

The proportion of patients with metastasis at diagnosis varied according to age group; the highest percentage was found in group 3 (47.4%; Figure 1). In the entire cohort, most tumors were found on the right side: 54.8%, 64.2%, and 65% in groups 1, 2, and 3, respectively. However, the tumor location (right or left) showed no significant association with the prognosis (*p* > 0.05). Moreover, tumor size ≥ 5 cm was found to be an important predictor of worse prognoses in groups 1 and 3, increasing the risk of developing metastasis by 5.3 times (*p* < 0.05; HR: 5.3; CI 95% 1.03–5.4; Figure 3).

### 3.5. Histological Data (Weiss/Modified Weiss Score and Wieneke Index) and Staging (MacFarlane and TNM)

All the groups with the following characteristics were strongly associated with malignant outcomes: tumor weight ≥ 200 g (*p* < 0.001; HR: 16.2; CI 95% 0.9–2.1; Table 3, Weiss/Modified Weiss score ≥ 5 (*p* < 0.05; HR: 6.2; CI 95% 1.2–8.3) and Wieneke index ≥ 3 (*p* < 0.001; HR: 25.8; CI 95% 1.4–3.4) (Table 4).

According to MacFarlane staging, patients with stage I and II who underwent complete surgical resection (R0) had the best prognosis in all groups (*p* < 0.001; HR: 5.1; CI 95% 1–25.9), whereas stage IV was associated with approximately two times shorter overall survival (OS) than stages I, II, and III (Figure 4).

### 3.6. Pediatric Adrenocortical Tumors Molecular Markers (Melan A, SF1, Inhibin, p53, and Ki67)

The results of immunohistochemical analysis were available for 60 out of the 95 patients (63.15%), and all these patients were positive for SF1 or inhibin and melan A.

Group 1 had much a higher percentage of patients (64.5%) with positive immunohistochemistry for p53 compared with groups 2 and 3 (28.6% and 10%, respectively); however, p53 showed no significant association with prognosis (*p* > 0.05).

Ki67 (an important immunohistochemical proliferation marker) was assessed in our cohort. The Ki67 immunostaining of ≥15% was related to high accuracies in the prediction of malignant behavior in all groups, increasing the risk of developing metastasis by 8 times (*p* < 0.05; HR: 8.1; CI 95% 1.2–8.2).

### 3.7. Genetic Analysis (TP53, BUB1B-PINK1, IGF-IR, ATRX, ZNRF3, MLH1, MSH6, and XAF1): Compiled Data from Previously Published Manuscript of This Cohort

We analyzed 61 patients for germline variants on TP53. The p.R337H was present in 76.7%; one of them had homozygous statuses (patient #4; attachment 4). This variant was highly prevalent, especially in the younger group, affecting 64.5%, 35.7%, and 20% of cases in groups 1, 2, and 3, respectively. For this variable, statistical analyses were applicable only for groups 1 and 3, since they could not be performed for group 2 due to the small number of patients with this event of interest. Nonetheless, the results showed no significant influences of this germline pathogenic variant on prognosis.

Furthermore, the difference in disease-free survival (DFS) using the cutoff of subtraction expression levels (BUB1B-PINK1) < 6.95 showed a marginally significant difference in developing metastasis (log-rank *p* = 0.062). This molecular predictor was not associated with a statically significant difference in the overall survival (OS) (log-rank *p* = 0.146) [21].

Insulin-like growth factor-II (IGF-II) is involved in the molecular pathogenesis of ACT. An overexpression of IGF-II transcripts in both pediatric ACC and ACA was observed (50.8 ± 18.5 and 31.2 ± 3.7, respectively; *p* = 0.23). Similarly, the overexpression of the IGF-II receptor (IGF-IR) was considered a risk factor for malignancy, as IGF-IR mRNA levels were significantly higher in children with PACTs exhibiting malignant evolution than with benign outcomes (9.1 ± 3.1 versus 2.6 ± 0.3, respectively, *p* = 0.01; HR: 1.84) [21].

Twelve patients with PACTs were evaluated for ATRX and ZNRF3. The ATRX pathogenic variants were found only in the group 1 (16.7%), whereas somatic ZNRF3 pathogenic variants were found in groups 1 and 2 (3.2% and 14.3%, respectively). Nevertheless, no significant clinical correlation with dismal evolutions was identified in these patients.

Moreover, 35 patients underwent MLH1 and MSH6 analyses (attachment 4). A high prevalence of pathogenic variants in MLH1 or MSH6 was observed to be associated with the germinative pathogenic variant of TP53 in patients with PACT (three out of 35), characterizing Lynch’s syndrome. Overall, three out of 35 patients presented germline pathogenic MMR genes (Attachment 4: 10#, 11#, and 81#) as the TP53 germline mutation (Arg337His), including two from group 1 (Attachment 4: 10# and 11#) and one from group 2 (Attachment 4: 81#). None of the patients presented with metastasis at diagnosis or during the follow-up period; however, patient 25# (Attachment 4) with MSH6 variant of uncertain significance developed metastasis during follow-up.

Of the 21 patients analyzed for the E134* variant in the XAF1 (attachment 4), the percentages of patients in groups 1, 2, and 3 who carried it were 24.2%, 14.3%, and 5%, respectively. However, this observation showed no significant association with prognoses in any of the groups.

### 3.8. Treatment

Seventy-one percent of patients underwent open surgical approaches; the vast majority (86.2%) had the tumor completely resected. Similar recurrence rates were observed between patients operated on by open surgery versus those operated on laparoscopically [22]. There was one case treated for local recurrence by cryoablation and immunotherapy (pembrolizumab; Table 3—patient #79).

Mitotane was used in all metastatic patients (*n* = 13), and it was used in association with chemotherapy (EDP+M) in metastatic patients for first-line treatments (*n* = 8); three patients are currently alive after undergoing chemotherapy.

The best treatment for normalizing clinical features regardless of hormonal excess is surgery. For recurrence cases, the control of tumor-related hypercortisolism was achieved by using mitotane. Hyperandrogenism was controlled by androgen receptor inhibitors; the GnRH analog was used to inhibit the development of central precocious puberty.

### 3.9. Outcome

Younger patients were less likely to show disease progression (Figure 1). In group 1, only one patient who initially had no metastases at diagnosis developed metastases during the follow-up period. In group 2, four patients who had nonmetastatic diseases at diagnosis developed distant metastasis. In group 3, three patients developed metastases during the follow-up period (Table 3).

A total of 18 deaths were recorded, including one death due to hypercortisolism, one death due to colon cancer, and 16 deaths due to advanced ACC: five patients from group 1 (8%), five patients from group 2 (35.7%), and six patients from group 3 (30%).

The median follow-up was 5.2 ± 3.1, 6.7 ± 2.8, and 12.6 ± 4.7 years, respectively, for groups 1, 2 and 3. As shown in Kaplan–Meier curves (Figure 4, Figure 5 and Figure 6), in group 1, the overall 5- and 10-year raw survival rates were 88%, with overall 5- and 10-year DFS rates at 78% and overall 5- and 10-year raw RFS rates at 100%. In group 2, the overall 5- and 10-year raw survival rates were 49%, with an overall 5-year DFS rate at 38% and overall 5- and 10-year raw RFS rates at 70% and 48%, respectively. Furthermore, in group 3, a drop from 60 to 51% was observed in the overall 5-year raw survival rate. Moreover, the overall 10-year raw survival rate was 51%; overall 5- and 10-year DFS rates were of 54%; overall 5- and 10- year raw RFS rates were 89% and 71%, respectively.

## 4. Discussion

The retrospective analysis of this study corroborates the findings of a recent meta-analysis involving a large cohort (1006 patients; age range: 0–18 years) that identified the following predictor factors of worse outcomes: age ≥ 4 years, cortisol-secreting tumors, no complete surgical resection, tumor volume ≥ 200 cm^3^, tumor weight ≥ 400 g, tumor maximum diameter ≥ 5 cm, and stage IV disease [23]. Interestingly, we observed that a tumor weight below 200 g was not a benign predictor factor once we observed deaths in seven out of 13 patients with low tumor weight. Otherwise, six out of 13 had a median tumor weight of around 492 g.

Besides the aforementioned prognostic factors, the following data presented a significant result associated with worse outcomes in our study: high 11-deoxycortisol levels, Weiss/modified Weiss score ≥ 5, Wieneke index ≥ 3, Ki67 ≥ 15%, the subtraction of expression *BUB1B-PINK1* < 6.95, and IGF-IR overexpression.

In contrast to the previously cited meta-analysis, the current study classified the cohort into three different age groups because, despite the biphasic age distribution, they were in early childhood (three years of age) and adolescence (12 years of age). It is known that the natural course of PACTs in early childhood is significantly different and favorable compared with adolescence, which exhibits similar malignant development as observed in adults [24,25,26,27,28,29]. Likewise, our results are consistent with several studies related to age that were published from different countries. Accordingly, there seems to be a consensus for worse prognoses if the patient is four years old at diagnosis [23,30,31].

PACTs usually manifest as early onset with classical clinical features related to precocious puberty because of tumor hormone secretions, with VS being the most common presentation, particularly in the youngest group, and it is triggered by the isolated secretion of androgens [27,32,33]. The frequency of PACTs was 70%, 35.7%, and 31.6% in groups 1, 2, and 3, respectively, confirming previous data [27,32,33]. Moreover, 25%–50% of cases had mixed tumor secretions of androgens and glucocorticoids [27,34]. Differently, in our cohort, the presence of hypercortisolism associated or not with virilization was not related to a worst prognosis.

PACTs rarely manifest as isolated hyperaldosteronism or estrogen secretion or even as non-functional tumors [27,35,36]. Feminizing tumors in adults have extremely poor prognoses [35], a fate that is not shared in pediatric patients; the two patients with FS in our cohort are alive, with a long DFS, OS, and RFS.

As an interesting clinical observation, we noted that all patients (group 1) who were tall for their chronological age at diagnosis (corresponding hormone secretions associated with peripheral precocious puberty) continued to show high growth rates despite the effectiveness of the surgical intervention and the absence of central precocious puberty (Figure 7). Whether androgynous effects during early childhood could be triggers for maintaining the growth rate curve without decreases or hormonal stimuli remains unclear.

Surgery is the mainstay of treatments in stages I–III and is the only therapy that unquestionably cures or prolongs survival significantly. The histopathological classification often does not reflect the tumor-related biological behavior caused by PACTs. Tumors with malignant biological characteristics may show benign evolution depending on the status of surgical resection (R0), tumor stage, and age (especially in children aged ≤ 3 years). The histological classification criteria are challenging when applied to children because the established criteria used in adults (Weiss score ≥ 3) often lack sensitivity and specificity in pediatric patients [3,35]. In the present study, the Weiss score and modified Weiss score of ≥5 were related to a predictor of malignant behavior (*p* < 0.001; HR: 6.2; CI 95% 1.2–8.3), and a score of ≤2 had a negative predictive value of 100% for the occurrence of metastasis; therefore, this scoring system, with some adaptations, may also be useful for pediatric patients [17].

Furthermore, a Wieneke index ≥ 3 showed a prognostic correlation with high sensitivity, which is suggestive of malignancy. These data are in accordance with previous studies [35,37], which showed high sensitivity (89%) and specificity (76%) for the diagnosis of ACC [1].

Upon a histopathological confirmation of PACT, it is necessary to stage the disease in order to determine the prognosis and to plan surveillance. The staging system that is widely used is the system introduced by MacFarlane and modified by Sullivan [38]. It classifies the tumor into four stages based on tumor size, lymph node involvement, local invasion, and metastatic disease. MacFarlane stages I and II had the best prognosis when a successful surgical resection was performed (R0), while stage IV was associated with approximately two times shorter OS than in stages I, II, and III (*p* < 0.001; HR: 5.1; CI 95% 1–25.9; Table 5; Attachment 3) [39,40,41]. In addition to stage IV, capsule spillage was associated with a short overall survival.

The Ki67 (MIB-1) immunostaining index is a strong prognostic factor and should be part of the histological classification of PACTs. In this study, three experienced pathologists revised the immunostaining of KI67 and concluded that the immunostaining of Ki67 ≥ 15% had predictive accuracies for biological behavior (*p* < 0.05; HR: 8.1; CI 95% 1.2–8.2), with an 8-fold higher risk of developing a worse prognosis. Surprisingly, Ki67 < 10% ruled out malignant biological behavior, but a large number of benign tumors also showed Ki67 ≥ 10% (49%). Conversely, a few cases with ≥ 20% showed benign biological behavior (11%), but some malignant cases were missed when using this cutoff value for risk prediction (22%). In the cohort described by Sandru F. et al. [36], a Weiss score ≥ 6 and Ki67 ≥ 15% showed important diagnostic values [36].

Interestingly, we demonstrated, for the first time, the prevalence (8.5%; 3/35) of heterozygous germline pathogenic variants in one of the genes involved in *MMR* (*MLH1, MSH2, MSH6,* and *PMS2*) in a cohort of pediatric patients who also harbored the germline pathogenic variant in the *TP53* (Figure 8). Therefore, pediatric patients with ACTs should be strongly assumed to be at genetic risk of Lynch’s syndrome. However, no difference in tumor behavior associated with the presence of these variants was identified. The main focus of this study was on the possibility of identifying the relatives of these index cases under risk in order to improve the care of families.

The tumor behavior of each age group is similar at OS, DFS, and RFS Kaplan–Meier curves. While group 1 obtained better curves, group 3 presented them as well compromised (Figure 4, Figure 5 and Figure 6). Additionally, DFS and RFS drop significantly in groups 2 and 3, suggesting that once patients survived beyond five years, their chances of dying from the disease diminished precipitously. This reinforces different hypotheses regarding the pathogenesis of childhood ACC: early childhood tumors arise from the fetal zone of the adrenal gland, whereas hormone changes have important influences during adolescence [41].

The rarity of PACTs is a major limitation in characterizing these tumors and an obstacle in pathologic interpretation and therapeutic decision-making. Furthermore, our study also has the following limitations: retrospective design, a small sample size of metastatic tumors, and the lack of an assessment of all variables in all patients.

## 5. Conclusions

Large-scale pan-genomic analyses have improved our understanding of genetic and epigenetic changes underlying the pathogenesis of PACTs. However, their prognostic values in clinical practice remain not fully defined. Despite the well-pointed-out limitations of our study, we hope that our findings may help identify patients with higher risks of aggressive disease courses who would benefit from adjuvant treatments. Although there are high frequencies of germline pathogenic variants on *TP53*, these are not correlated with poor prognoses. In addition, whether *MMR* genes play a role in pediatric adrenocortical tumorigenesis and whether this association with the *TP53* variant influences the tumor’s behavior remain unclear, but there are potentially broad implications for clinical surveillance and genetic counseling for patients and their families.

## Figures and Tables

**Figure 1 jcm-11-06641-f001:**
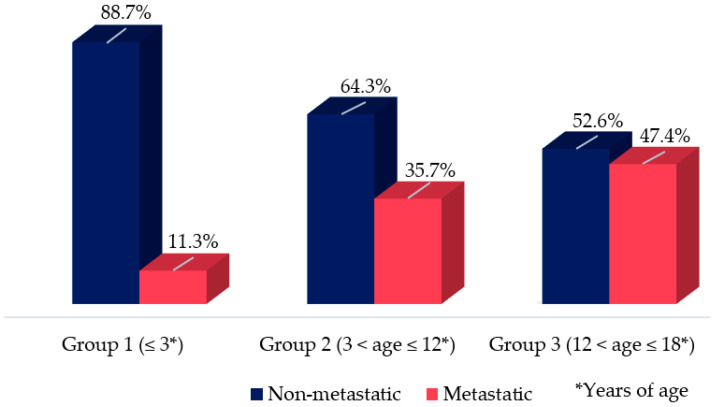
Percentage of metastatic and non-metastatic pediatric patients diagnosed with adrenocortical tumor according to age-groups; group 1: ≤3 years of age; group 2: 3 < age ≤ 12 years of age; group 3: 12 < age ≤ 18 years of age. Patient aged > 3 years are more likely to develop a malignant evolution (*p* = 0.03; HR: 2.5; CI 95% 1.07–5.8).

**Figure 2 jcm-11-06641-f002:**
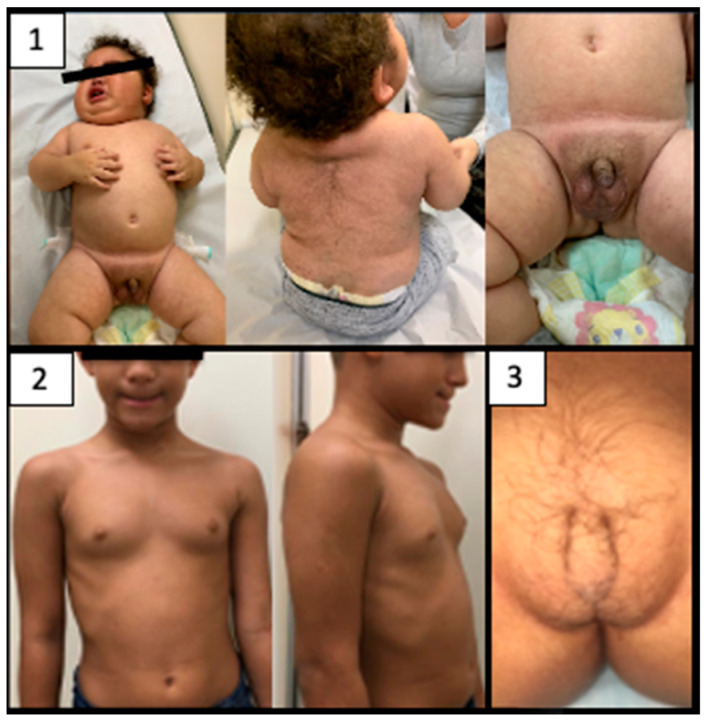
Clinical manifestations of patients with PACTs. 1: male patient (9 months of age) with mixed syndrome (VS and CS); 2: Male patient (5 years of age) with estrogen, androgen and cortisol producing tumor (gynecomastia and an important bone age advance); 3: Female patient (1 year of age) with pubic hair and clitoromegaly (2.7 cm). The parents gave a written informed consent for the publication of the images.

**Figure 3 jcm-11-06641-f003:**
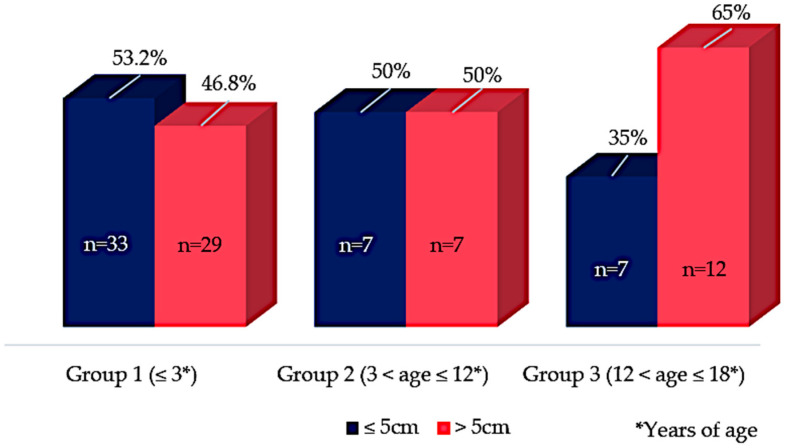
Tumor size and their relative frequencies (%) in each age-group (*n* = number of cases); group 1: ≤3 years of age; group 2: 3 < age ≤ 12 years of age; group 3: 12 < age ≤ 18 years of age. Tumor size ≥ 5 cm was found to be an important predictor of worse prognosis in groups 1 and 3, (*p* < 0.05; HR: 5.3; CI 95% 1.03–5.4).

**Figure 4 jcm-11-06641-f004:**
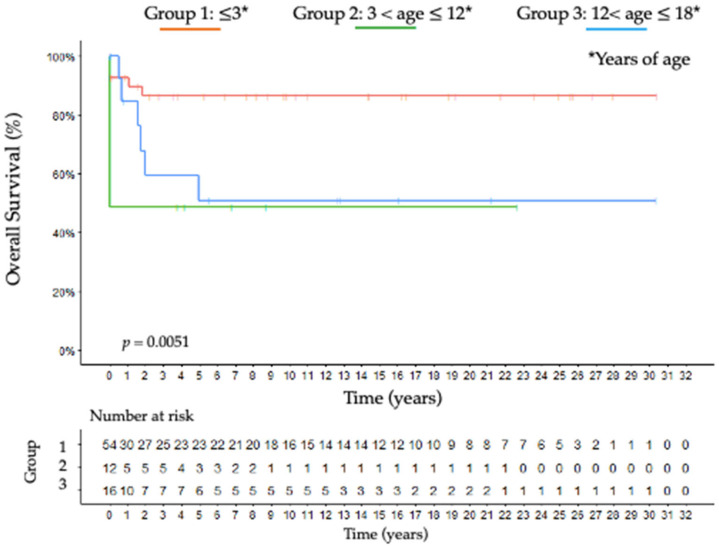
Kaplan-Meier curve estimates for overall survival according to each age-group; group 1: ≤3 years of age; group 2: 3 < age ≤ 12 years of age; group 3: 12 < age ≤ 18 years of age.

**Figure 5 jcm-11-06641-f005:**
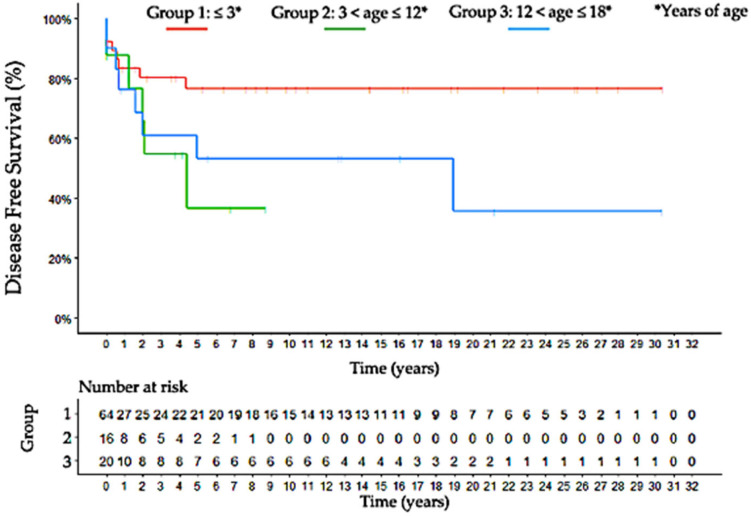
Kaplan-Meier curve estimates for disease-free survival according to each age-group; group 1: ≤3 years of age; group 2: 3 < age ≤ 12 years of age; group 3: 12 < age ≤ 18 years of age.

**Figure 6 jcm-11-06641-f006:**
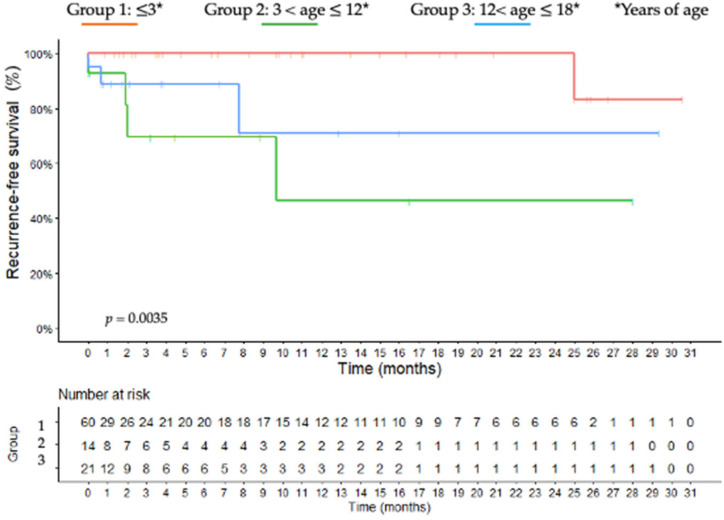
Kaplan-Meier curve estimates for recurrence-free survival according to each age-group; group 1: ≤3 years of age; group 2: 3 < age ≤ 12 years of age; group 3: 12 < age ≤ 18 years of age.

**Figure 7 jcm-11-06641-f007:**
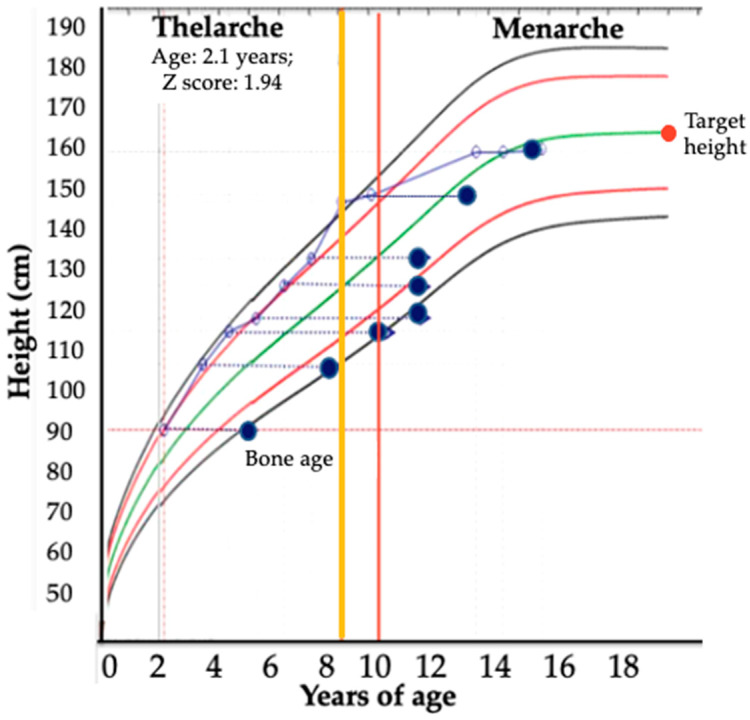
Growth pattern of male patient with PACT, 2 years of age, with mixed syndrome (V + CS), tumor weight of 250 g, sized 12 cm, McFarlane IV, Weiss score 7, Wieneke 4, with TP53- p.R337H and overall survival of 11 months. Blue circle is stature for age and red circle is bone-age.

**Figure 8 jcm-11-06641-f008:**
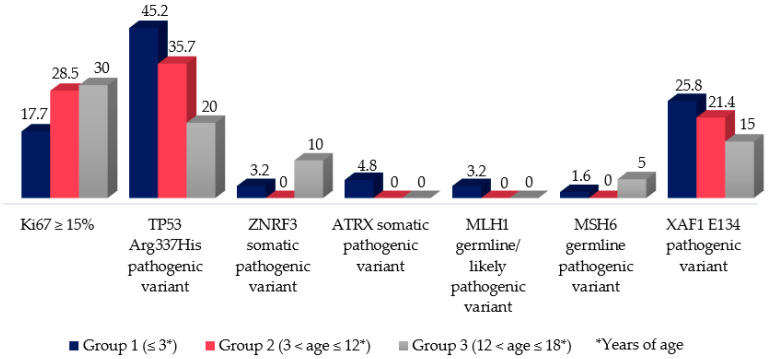
Percentage of patients in each age-groups in the evaluation of pediatric adrenal tumor-related genes; group 1: ≤3 years of age; group 2: 3 < age ≤ 12 years of age; group 3: 12 < age ≤ 18 years of age. Ki67 ≥ 15% increased the risk of developing metastasis in all groups (*p* < 0.05; HR: 8.1; CI 95% 1.2–8.2). The TP53 Arg337His pathogenic variant, ZNRF3/ ATRX somatic pathogenic variant, MLH1 germline/ likely pathogenic variant, MSH6 germline pathogenic variant and XAF1 E134 pathogenic variant were not associated with a worse outcome, in all groups (*p* > 0.05).

**Table 1 jcm-11-06641-t001:** Epidemiologic Features and their Relative Frequencies in Patients with PACT Age-Groups.

Group	No of Patients *n* (%)	Mean Age at Diagnosis (Years of Age)	Female (%)	Male (%)	From South/Southeast Regions of Brazil (*n*)	From Midwest Region of Brazil (*n*)	From Northeast Region of Brazil (*n*)	From Bolivia and United States (*n*)
1	62 (68.4)	1.9	45	17	46	10	5	1
2	14 (16.8)	7	10	4	12	1	0	1
3	20 (21)	17.3	17	3	12	3	3	0
		*p* = 0.03 (>3 years of age)		*p* = 0.09				
		HR: 2.5; CI 95% 1.07–5.8						

PACT: Pediatric Adrenocortical Tumor; No: number; Yrs: years; group 1: ≤3 years of age; group 2: 3 < age ≤ 12 years of age; group 3: 12 < age ≤ 18 years of age.

**Table 2 jcm-11-06641-t002:** Hormonal data of patients with PACTs from three age-groups.

Group	11-Deoxycortisol (ng/mL)	17-OHP (ng/mL)	SDHEA (ng/mL)	Testosterone (ng/dL)	Free Testosterone (pmol/L)	Estradiol (pg/mL)	LH (UI/L)	Cortisol (µg/dL)
1	6 ± 8 (<0.5)	15.3 ± 54.6 (<0.86)	23614.2 ± 46241.2 (<1240)	408.4 ± 775 (female: <14; male: <14)	194.4 ± 266.5 (female: 2.4–37; male: 131–640)	31.55 ± 52.2 (female: <21; male: <20)	0.81 ± 1.88 (female: 2.4–12.6; male: 1.7–8.6)	17.45 ± 53.48 (in the morning: 6.7–22.6)
2	8 ± 7.9 (<0.5)	3.2 ± 1.8 (<0.86)	7429.5 ± 9264.7 (<852)	363.6 ± 267.8 (female: <14; male: 3–32)	193 ± 172.5 (female: 2.4–37; male: 131–640)	27 ± 8.75 (female: 6–27; male: 25.8–60.7)	0.23 ± 0.21 (female: 2.4–12.6; male: 1.7–8.6)	12.96 ± 7.79 (in the morning: 6.7–22.6)
3	257.3 ± 392.9 (<45)	3.7 ± 3.3 (0.6–3.3)	5834.3 ± 8493.2 (678–4127)	395.4 ± 267.8 (female: <98; male: 200–830)	132.7 ± 183.6 (female: 2.4–37; male: 131–640)	45.81 ± 51.52 (female: follicular phase: 19–247; luteal phase 22–256; male: <56)	4.17 ± 9.93 (female, follicular phase: 20–125; luteal phase: 5–17; male: 1.4–9.2)	18.98 ± 4.85 (in the morning: 4–22)
	*p* < 0.001	*p* = 0.297	*p* = 0.416	*p* = 0.518	*p* = 0.511	*p* = 0.501	*p* = 0.477	*p* = 0.263
	HR: 2.9 CI 95% 0.9–1.2							

PACT: pediatric adrenocortical tumor; The normal range are in parentheses; group 1: ≤3 years of age; group 2: 3 < age ≤ 12 years of age; group 3: 12 < age ≤ 18 years of age.

**Table 3 jcm-11-06641-t003:** Characteristics of patients, initially non-metastatic at diagnosis that developed metastasis during the follow-up (#44; #60; #63; #66; #73; #88; #92), metastatic patients at diagnosis (#4; #30; #55; #56–#59; #64; #71; #74; #79; #83; #89–#91; #93; #94) and non-metastatic (#1–#3; #5–#29; #31–#43; #45–#54; #61; #62; #65; #67–#70; #72; #75–#78; #80–#82; #84–#87; #95).

Patients	Gender	Age (yrs)	Clinical	Tu Size (cm)	Weight (g)	Wieneke Index	Weiss Score	M. Weiss Score	Ki67 (%)	p53 p.R337H	MacF. Stage	11-Deoxycortisol (<0.5 ng/mL)	Decease
	*p* = 0.09	*p* = 0.03 (>3 yrs of age)	*p* = 0.057	*p* < 0.05 (≥5 cm)	*p* < 0.001 (≥200 g)	*p* < 0.001 (≥3)	*p* = 0.002 (≥5)	*p* < 0.001 (≥5)	*p* < 0.05 (≥15%)	*p* > 0.05	*p* < 0.001 (I–III)	*p* < 0.001	
		HR: 2.5; CI 95% 1.07–5.8	HR 0.042; CI 95% 0.002–1.1	HR: 5.3; CI 95% 1.03–5.4	HR: 16.2; CI 95% 0.9–2.1;	HR: 25.8; CI 95% 1.4–3.4	HR: 6.2; CI 95% 1.2–8.3	HR: 6.2; CI 95% 1.2–8.3	HR: 8.1; CI 95% 1.2–8.2		HR: 5.1; CI 95% 1–25.9	HR: 2.9; CI 95% 0.9–1.2	
#1	F	0.1	V + C	3.9	31.4	1	7	5	50	yes	I	3.17	No
#2	F	0.1	V	6.4	47	3	8	6	30	yes	II	0.4	No
#3	M	0.7	V	4	n.a.	3	6	6	43	yes	I	n.a.	No
#4	F	0.8	V	4	25	3	6	5	40	yes	IV	n.a.	No
#5	F	0.11	V	4.5	31.3	1	3	3	13	no	I	1.8	No
#6	F	0.11	V	6.5	112	3	6	6	16	yes	II	6.6	No
#7	F	0.11	V + C	8.5	138	3	4	3	n.a.	yes	II	n.a.	No
#8	F	1.11	V	5	45	1	2	1	11	no	I	n.a.	No
#9	F	0.8	V + C	3.5	50	2	6	6	31	yes	I	n.a.	No
#10	M	0.9	V + C	8.5	89	4	9	7	10	yes	III	2.2	Yes
#11	F	1	V + C	4.5	n.a.	1	1	2	n.a.	no	I	n.a.	No
#12	M	1	V + C	11	n.a.	3	5	4	n.a.	no	II	n.a.	No
#13	F	1	V	4.5	20	n.a.	2	n.a.	n.a.	no	I	n.a.	No
#14	F	1	V + C	6	60	n.a.	n.a.	n.a.	n.a.	no	III	n.a.	No
#15	F	1	V + C	6.4	60	1	2	2	15	yes	II	n.a.	No
#16	F	1	V	3.3	11.8	n.a.	2	n.a.	n.a.	yes	I	n.a.	No
#17	F	1	V	7	100	1	1	1	n.a.	yes	II	n.a.	No
#18	F	1.1	V	5	190	2	6	4	1	no	III	n.a.	No
#19	M	1.1	V	6	n.a.	3	6	5	31	no	III	n.a.	No
#20	F	1.3	V + C	3.7	10	2	5	5	6	yes	I	n.a.	No
#21	F	1.3	V	5	40	1	1	1	n.a.	no	I	n.a.	No
#22	M	1.4	V + C	5	40	1	7	2	16	yes	I	n.a.	No
#23	M	1.4	V	5	30	1	2	1	2	yes	I	n.a.	No
#24	M	1.7	V	1.5	4	2	5	5	1	yes	I	n.a.	No
#25	F	1.8	V	9.5	140	2	5	5	4	no	II	n.a.	No
#26	F	1.9	V	3	10	3	6	6	2	no	I	n.a.	No
#27	M	1.9	V	5.5	55	2	4	1	1	no	II	n.a.	No
#28	F	1.9	V + C	7.5	38	2	5	1	n.a.	yes	III	3.9	No
#29	M	2	V	3.5	10	4	4	5	11	yes	I	n.a.	Yes
#30	M	2	V + C	12	250	6	7	6	19	yes	IV	n.a.	Yes
#31	F	2	V	3	n.a.	1	1	1	n.a.	yes	I	n.a.	Yes
#32	F	2.1	n.a.	7	n.a.	5	8	7	n.a.	yes	II	n.a.	No
#33	F	2.1	V	5	30	1	3	2	5	yes	I	n.a.	No
#34	F	2.1	V	4	20	1	3	2	13	yes	I	n.a.	No
#35	M	2.1	V	6	135	3	5	5	2	yes	II	n.a.	No
#36	F	2.2	V	3	5	2	2	2	10	yes	I	n.a.	No
#37	M	2.2	V	6.5	90	2	4	2	8	yes	II	n.a.	No
#38	F	2.3	V	7	55	2	2	1	2	yes	II	n.a.	No
#39	M	2.3	V	4.5	10.7	3	6	6	18	yes	I	n.a.	No
#40	F	2.6	V	3.8	5	1	5	5	13	yes	I	6.1	No
#41	F	2.6	V	5.2	55	4	7	7	2	yes	II	n.a.	No
#42	F	2.6	V	4.5	10	1	1	2	1	yes	I	6.3	No
#43	F	2.6	V	2.5	5	2	4	5	10	no	I	n.a.	No
#44	M	2.7	V	2.5	3	3	5	5	40	yes	I	4	No
#45	M	2.7	V	2.5	3	3	5	5	40	yes	I	4	No
#46	F	2.8	V	5.5	40	3	6	6	10	yes	II	n.a.	No
#47	F	2.9	V	1.8	4	1	3	1	3	yes	I	5.4	No
#48	F	3	V + C	6.5	48.5	1	5	5	13	yes	II	0.49	No
#49	F	3	V	3	n.a.	1	3	1	n.a.	no	I	n.a.	Yes
#50	F	3	V + C	4.5	20	3	3	4	n.a.	yes	I	n.a.	No
#51	M	3	V	5.5	8.2	1	4	1	10	yes	I	0.24	No
#52	F	3	V	2.5	7.3	1	7	5	10	yes	I	2.1	No
#53	F	3	V + C	6	37.9	4	5	3	40	yes	II	31.8	No
#54	M	3	V + C	5.5	30	2	4	4	40	no	I	n.a.	No
#55	M	3	V	11	381	8	9	7	n.a.	n.a.	IV	n.a.	Yes
#56	F	3	V + C	6	70	n.a.	n.a.	n.a.	n.a.	n.a.	IV	n.a.	No
#57	M	3	V + C	12	970	6	8	7	40	yes	III	7.47	No
#58	F	3.11	V + C	13.5	650	8	7	7	27	Yes	IV	n.a.	No
#59	M	3.3	V	6	70	n.a.	7	n.a.	n.a.	Yes	IV	17	No
#60	M	3.3	V + C	6	37.9	4	5	3	40	Yes	III	17	No
#61	F	3.4	V	1.5	3.2	1	3	2	5	Yes	I	0.7	No
#62	F	3.8	V	1.2	3	1	1	1	n.a.	no	I	n.a.	No
#63	M	4	V + C	6	70	5	7	5	57	Yes	II	n.a.	No
#64	M	4	V + C	6	180	2	4	3	n.a.	n.a.	IV	n.a.	Yes
#65	F	4.6	V + C	6.5	70	3	5	4	57	No	II	n.a.	Yes
#66	F	4.6	V + C	6.5	72	3	5	4	52	Yes	III	n.a.	No
#67	M	5	V + F + C	4.5	34.4	6	3	2	n.a.	Yes	I	12.7	No
#68	F	6	V	7.5	35	3	5	3	10	No	II	n.a.	No
#69	F	6	V	4	n.a.	2	6	6	25	Yes	I	n.a.	No
#70	F	6	n.a.	1	5	n.a.	n.a.	n.a.	n.a.	No	II	n.a.	No
#71	F	6.2	V + C	7	55	5	8	6	11	No	IV	n.a.	Yes
#72	F	7	V + C	n.a.	n.a.	0	0	0	n.a.	No	n.a.	n.a.	Yes
#73	F	7	V + C	n.a.	n.a.	n.a.	n.a.	n.a.	n.a.	n.a.	n.a.	n.a.	No
#74	F	9.6	C	6	220	5	7	7	26	No	IV	n.a.	Yes
#75	M	9.6	V + C	4.5	20	0	2	1	1	No	I	n.a.	No
#76	F	10	V + C	5	20	3	6	6	n.a.	No	II	n.a.	No
#77	M	11.5	V + C	2.5	45.6	0	2	2	n.a.	Yes	I	16.8	No
#78	F	13	V	6.5	51	0	0	0	n.a.	No	II	n.a.	No
#79	M	14	V + C	n.a.	n.a.	n.a.	n.a.	n.a.	n.a.	n.a.	n.a.	n.a.	No
#80	F	14.5	C	2	n.a.	0	1	0	n.a.	No	I	n.a.	No
#81	F	15	C	3.2	20	0	2	1	n.a.	No	III	n.a.	No
#82	F	15.7	C	7	60	1	2	3	n.a.	No	II	n.a.	No
#83	F	16	V + C	20	1475	8	7	6	n.a.	Yes	IV	24	Yes
#84	F	16.11	V + C	4	5	0	2	1	3	No	IV	n.a.	No
#85	M	17	V	9	165	5	7	3	n.a.	No	II	37	No
#86	M	17	V	5	29	2	5	5	7	No	I	n.a.	No
#87	F	17	V	20	825	8	8	5	n.a.	No	III	n.a.	Yes
#88	F	17.3	V	24	725	7	8	7	n.a.	No	III	2.7	Yes
#89	F	17.4	C	10	5	5	8	7	40	Yes	III	711	Yes
#90	F	18	V	13	1000	8	8	7	n.a.	Yes	IV	n.a.	Yes
#91	F	18	V	12	280	6	8	7	n.a.	n.a.	IV	n.a.	Yes
#92	F	18	C	2.6	15	2	4	3	n.a.	n.a.	I	n.a.	No
#93	F	18	n.a.	16.5	n.a.	n.a.	6	4	n.a.	n.a.	IV	n.a.	No
#94	F	18	V + C	12.5	n.a.	n.a.	n.a.	n.a.	n.a.	n.a.	IV	n.a.	Yes
#95	F	18	C	2.6	15	2	4	3	n.a.	Yes	I	n.a.	No

Yrs: years; Tu: tumor; M. Weiss score: Modified Weiss score; MacF. Stage: MacFarlane Stage; M: male; F: female; Clinical presentation: V- virilization syndrome; C- Cushing’s syndrome; F: Feminizing syndrome; n.a.: not available; group 1: ≤3 years of age; group 2: 3 < age ≤ 12 years of age; group 3: 12 < age ≤ 18 years of age.

**Table 4 jcm-11-06641-t004:** Percental of the different staging of patients with PACTs divided in three different age-groups.

Group	Weiss Score ≥ 5	M. Weiss Score ≥ 5	Wieneke Index ≥ 3	MacFarlane Stage ≥ II	TNM Stage ≥ II
1	35.5%	24.2%	17.7%	21%	14.5%
2	50%	28.6%	28.6%	28.6%	28.6%
3	55%	50%	55%	40%	40%
	*p* = 0.002	*p* = 0.018	*p* < 0.001	*p* < 0.001	*p* < 0.001
	HR: 6.2; CI 95% 1.2–8.3	HR: 6.2; CI 95% 1.2–8.3	HR: 25.8; CI 95% 1.4–3.4	HR: 5.1; CI 95% 1–25.9	HR: 5.1; CI 95% 1–25.9

group 1: ≤3 years of age; group 2: 3 < age ≤ 12 years of age; group 3: 12 < age ≤ 18 years of age; M. Weiss score: Modified Weiss Score.

**Table 5 jcm-11-06641-t005:** Difference among the staging systems TNM, MacFarlane and MacFarlane/Sullivan.

Stage	TNM	MacFarlane	MacFarlane/Sullivan
I	T1, N0, M0	T1, N0, M0	T1, N0, M0
II	T2, N0, M0	T2, N0, M0	T2, N0, M0
III	T1-2, N1, M0; T3, N0, M0	T3, N0, M0 T1-3, N1, M0 (mobile nodes)	T3, N0, M0 T1-3, N1, M0
IV	T3, N1, M0; T4, N0-1, M0; T1-4, N0-1, M1	Any T, N1 or M1 (fixed nodes)	T4, N1, M0 or any M1

T1: diameter tumor < 5 cm, contained in the adrenal gland; T2: diameter tumor > 5 cm, not contained in the adrenal gland; T3: tumor extends beyond the fat capsule of the adrenal gland; and T4: tumor has extended to adjacent organs. N0: absence of compromised lymph node; and N1: the tumor has spread to nearby lymph nodes. M0: absence of distant metastasis; and, M1: presence of distant metastasis.

## Data Availability

Not applicable.

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
