# Peer review of "Retrospective Analysis of Prognostic Factors in Pediatric Patients with Adrenocortical Tumor from Unique Tertiary Center with Long-Term Follow-Up"

_jcm, 2022, doi:10.3390/jcm11226641_

Round 1

Reviewer 1 Report

The manuscript by Bachega et al. is a retrospective analysis of 95 patients with adrenocortical tumors observed and treated at one single Institution in Brazil. The manuscript comes from a group, leaded by prof Fragoso, that has a very well known scientific international reputation in the field of pediatric endocrinological diseases and particularly of pediatric adrenal tumors.

The work poses the important issue of analysis of clinical, pathological and genetic factors associated with metastasis development and prognosis in pediadric patients with adrenal tumors.

Overall, I think that this work has many merits. At the same time, I think that some points require some attention and should be revised.

Minor points:

·      Language/mistake revision is needed: there are some mistakes through the text (eg Title of Table 2 [line 335]: with your respectively standard deviation ?)

Major points:

·      The declared endpoint of this study is an analysis of prognostic factors (Introduction: lines 71-76) in pediatric adrenal tumors. This analysis included a large number of features ranging from clinical presentation, hormonal hyperfunction, pathologic features and genetic markers. The first concern is if this study has enough sample size and statistical power to detect differences in all groups. As the authors correctly state (lines 193-194), due to small sample size the multivariate model was not adjusted. However, I could not find any result of hazard ratio from univariate Cox analysis of prognostic factors which should be the main result of the study.

·      In line with this, in the Outcome sub-paragraph there is a Kaplan-Meier curve but there is no mention of the median follow-up and median survival outcomes (DFS, PFS, OS). I think it would be important and interesting to have this information.

·      This study covered a long time-frame. How were these patients treated along this period? How many received surgery? What about mitotane? Hypersecretion treatments? The authors recently published in the JCO a study of pediatric ACC treated with surgery and EDP chemotherapy. In the present series, did patients also received chemo?

·      This series included both adrenocortical carcinomas and adenomas. Although the distinction in pediatric patients may be not so clear-cut, I would like to stratify patients according to this criteria and then according to stage (metastatic vs no at diagnosis), syndromic vs sporadic tumors.

·      The genetic part of the study comes from a previous publication. It is not clear to me whether this genetic analysis was based on germline DNA, somatic DNA or both. Would it be possible to better specify? Were the analyzed genes part of a prespecified panel?  If it was a germline analysis (as in the case of TP53), was there a correlation with familial history of cancer or Li-Fraumeni syndrome or Lynch syndrome or other conditions? In the materials and methods section the description of lab methods is very long (from line 118 to 184) and it would be better to reduce it in sub-paragraph or move part of it in a supplementary section.

·      Did the authors make any correlation study between categorical or continuous variables by means of parametric or non-parametric tests? Was a given group characterized by association with clinico-pathological  or genetic characteristics in a statistically significant way? The bar-graphs that are reported in the text are descriptive but less informative on association p values.

·      I appreciated that the authors disclosed the limitations of their study. These can be found in different sections of the manuscript and perhaps, they could be all described at the end in the Conclusions section.

·      In conclusion, I think that a series of around 100 patients with PACT does indeed contain important information for the scientific community. However, in its present form this study represents more a case-series description than a quantitative analysis of prognostic factors and I think that a better structure/analysis of existing data could really improve the quality of this work. If for any reason, this cannot be done (missing data, no follow-up, etc.) then I would suggest to state this in the Introduction section.

Reviewer 2 Report

Overall, a very nice work on an interesting topic. Unfortunately, the graphics are a little blurred. In addition, the colours of the figures are a bit too bright, you should change the yellow in the figures. Also, in one of the figures the bar charts are arranged diagonally, this should be arranged horizontally. 

Reviewer 3 Report

Hello, thank you for your interesting work! I respect the efforts you put into the manuscript. 

Major aspects: 

1. Please have a native english speaker revise the manuscript. 

2. Please include a "basic patient characteristics" table in your manuscript. 

3. Please state more information on your statistical prognostic analysis; for example, what does the following mean?: "For this variable, statistical analysis was applicable only for groups 1 281 and 3; the results showed no significant influence of this mutation on prognosis." Hazard Ratios are stated throughout the text without clear overview of your findings.  

Minor aspects: 

1. Your age groups are mathematically incorrect: Group 1 <= 3, Group 2 4 <= age <= 12, group 3 13 <= age <= 18; ages between 3 and 4 as well as 12 and 13 aren't covered. 

2. In the material and methods part, you already state results: "The mean age of all patients at the time of diagnosis was 5.8 years (range 0.58 –18 93 years); 71.5% were female (F:M ratio 2.5:1). The follow-up ranged from 27 to 408 months." The same happens in Conclusions: "Furthermore, Wieneke index 3 3 showed a prognostic correlation with high sensitiv- 400 ity, suggestive of malignancy (p < 0.001; HR: 2), this data is in accordance with previous 401 studies [26,28], that showed sensitivity (89%) and specificity (76%) for the diagnosis of 402 ACC [1]."

3. Please state the actual result in: "Although 205 the proportion of girls was higher than boys in all groups, sex showed no significant as- 206 sociation with prognosis (p > 0.05)."; what was the actual p value, what was the Odds Ratio? This applies to several other phrases like "Patients aged 3 3 years showed significantly worse clinical outcome (p < 0.05)."; in what exact way was it worse and what was the p value? The above named are only examples and this "error" occurs at multiple sites throughout the manuscript. 

4. Please state p values for comparison between groups in your tables. 

Round 2

Reviewer 1 Report

The Authors have effectively answered the points I have raised. I think the manuscript in quality over its initial version.

One minor point: when reporting Hazard ratios in univariate Cox analysis they add the "IC95%". Does it stand for Confidence Interval 95%?

I found on Wikipedia a different definition of Information Coefficient = The information coefficient (IC) is a measure of the merit of a predicted value. In finance, the information coefficient is used as a performance metric for the predictive skill of a financial analyst

Could you please check? I think the correct meaning would be Confidence Intervals 95%. 

Reviewer 3 Report

Dear colleagues,

I appreciate the extra work you put into your manuscript. In my opinion, your revisions entailed an improvement to your work , however, you must have it revised with regards to language by a native english speaker. 
